# A Current Review of the Etiology, Clinical Features, and Diagnosis of Urinary Tract Infection in Renal Transplant Patients

**DOI:** 10.3390/diagnostics11081456

**Published:** 2021-08-12

**Authors:** María Luisa Suárez Fernández, Natalia Ridao Cano, Lucia Álvarez Santamarta, María Gago Fraile, Olivia Blake, Carmen Díaz Corte

**Affiliations:** 1Unidad de Gestión Clínica de Nefrología, Hospital Universitario Central de Asturias, 33011 Oviedo, Spain; nataliaridaocano@yahoo.es (N.R.C.); lucia.alvarez@sespa.es (L.Á.S.); mgagofraile@gmail.com (M.G.F.); carmen.diazc@sespa.es (C.D.C.); 2Universidad de Oviedo, 33003 Oviedo, Spain; olivialblake@hotmail.com

**Keywords:** urinary tract infection, kidney transplantation, asymptomatic bacteriuria, cystitis, pyelonephritis, recurrent urinary tract infections

## Abstract

Urinary tract infection (UTI) represents the most common infection after kidney transplantation and remains a major cause of morbidity and mortality in kidney transplant (KT) recipients, with a potential impact on graft survival. UTIs after KT are usually caused by Gram-negative microorganisms. Other pathogens which are uncommon in the general population should be considered in KT patients, especially BK virus since an early diagnosis is necessary to improve the prognosis. UTIs following kidney transplantation are classified into acute simple cystitis, acute pyelonephritis/complicated UTI, and recurrent UTI, due to their different clinical presentation, prognosis, and management. Asymptomatic bacteriuria (ASB) represents a frequent finding after kidney transplantation, but ASB is considered to be a separate entity apart from UTI since it is not necessarily a disease state. In fact, current guidelines do not recommend routine screening and treatment of ASB in KT patients, since a beneficial effect has not been shown. Harmful effects such as the development of multidrug-resistant (MDR) bacteria and a higher incidence of *Clostridium difficile* diarrhea have been associated with the antibiotic treatment of ASB.

## 1. Introduction

Urinary tract infections (UTIs) are the most common infection in kidney transplant (KT) patients, accounting for 42–75% of overall and 30% of all hospitalizations for sepsis [1,2,3]. The prevalence of UTIs varies between the series from 7 to 80%, due to the differences in definition, prophylactic antibiotic regimens, geographical location, and follow-up periods [4]. A meta-analysis that included 13 studies with a total of 3364 patients, found a pooled prevalence of UTIs of 38% with a range from 16 to 75%, and there were significant differences between Americans and Europeans by subgroup analysis (41% vs. 33%) and according to the follow-up time, being higher in patients followed for 1–2 years than those followed for 2–5 years (34% vs. 43%) [5]. In addition to this high incidence, UTIs in KT recipients are a major cause of morbidity, hospitalization, and mortality, with impact on the patient’s well-being and associated with other complications such as, development of resistant bacteria, drug interactions, and potential effects on graft survival. Therefore, diagnosis, prevention, and management of UTI, as well as the understanding of the special predisposition of these patients, are essential in the KT practice in order to minimize their complications. This article reviews definitions, etiology, clinical criteria, diagnosis and screening of UTIs in KTs, as well as their risk factors and impact on graft survival.

## 2. Risk Factors

The risk factors for UTIs in KT recipients are the same as in the general population, such as female gender, elder age, pre-transplant urinary tract abnormalities, diabetes mellitus, a history of recurrent UTIs or polycystic kidney disease [1]. Female gender is a well-known risk factor due to the anatomical differences of the urinary tract. However, in some studies, similar rates of incidence have been found in men and women in the first 6 months after KT, while by 3 years post-transplant, there was a higher incidence in women compared with men [5,6]. The incidence of UTIs increases with age [5,6,7,8] and it may be due to an impaired immunological system, higher rate of urinary retention secondary to bladder atrophy and/or prostatism and more complicated postoperative management in elderly patients. On the other hand, some authors have not found a correlation between age and UTI in KT patients [9,10].

In addition, other specific risk factors are present in KT recipients, which explain the greater incidence of UTI compared with the general population and non-renal transplants. In the RESITRA Spanish registry of more than 4000 solid organ transplants (SOT), the overall incidence rate of UTIs was 0.23 episodes per 1000 transplantation days, while the incidence rate in KT was 0.45 episodes per 1000 transplantation days and all cases of pyelonephritis occurred in KT patients [7]. These differences are explained by multiple factors that include surgical, anatomical, recipient, and graft-related factors, as well as, immunosuppression status. Prolonged dialysis prior to transplantation is associated with oliguria in most patients, which can lead to a disused and dysfunctional bladder that may regain function over time and with improved urine flow after transplantation. In this period of recovery, the bladder may be unstable and experience increased pressure of the detrusor muscle, predisposing to higher urinary frequency, urgency, and reflux into the transplant or native systems, which may increase the risk of UTI, especially if the urine is not sterile [11]. Vesico-ureteric reflux (VUR) increases the pyelonephritis risk, which is common in KT since there are several factors that predispose to it. Some of these factors are the lower localization of the graft, its proximity to the bladder, and a shorter ureter with a short intramural component in the bladder with a Lich-Gregoir technique, which is the recommended by the guidelines [11]. Furthermore, the presence of the temporary ureteric stent also favors VUR, abolishing peristalsis and dilating the vesico-ureteric junction [12]. The routine use of a prophylactic transplant ureteric stent is recommended since it reduces major urological complications [13] but is associated with increased UTI risk, which may be lower with stent removal before 14–21 days [14,15]. The use of urethral catheters also increases the risk of bacteriuria by 5% per day in the general population [16] as it perturbs the host’s defense mechanisms and provides easier access to the bladder by uropathogens. In the transplant population, the UTI rates in the first month can increase by 33% with a catheter left in for 9 days [17], therefore early catheter removal, traditionally on the fifth day after surgery, is advised to reduce infections and hospital stays. Some studies have even advocated for catheter removal as soon as the first or second day demonstrating evidence of a decrease in UTI rates [18,19]. KT from deceased donors and expanded criteria donors are associated with an increased susceptibility of UTIs, while living donor transplantation is associated with a lower incidence [5,10,17,20], probably since they are subjected to shorter periods of cold ischemia and less severe ischemic-reperfusion injury with a lower rate of delayed graft function, which is a predisposing factor for the development of UTIs [17,20]. Acute rejection has also been linked with a higher risk of UTIs [5,21] and the reason may be that its treatment requires more intense immune system suppression.

Finally, immunosuppression in KT recipients contributes strongly to the risk for infection, including UTI. The immunosuppressive drugs that have been shown to be associated with a higher rate of UTIs, have been anti-thymocyte globulin [22,23], azathioprine [24], and mycophenolate mofetil [25], while other drugs (calcineurin inhibitors, m-Tor inhibitors) [1] or steroid withdrawal [26] have not shown any effect on the risk of UTI. Denosumab, although not considered to be immunosuppressive, is a monoclonal antibody used to prevent bone loss and is associated with an increased risk of UTIs (however, only cystitis, neither pyelonephritis nor urosepsis) but not with other transplant specific infections [27]. Sodium glucose cotransporter-2 (SGLT2) inhibition, an emerging therapeutic option in treatment of diabetes mellitus, have been shown to reduce cardiovascular mortality and preserve kidney function in patients with type 2 diabetes. All these benefits make them an interesting option in KT patients but the potential risk of UTI due to glycosuria limits its use, especially if the history of recurrent UTIs is present [28]. However, these agents (empagliflozine and canagliflozine) have shown a low risk of UTI in KT recipients with no history of recurrent UTIs in a few small studies [29,30,31]. In addition, the use of loop diuretics in KT recipients has also been associated with an increased rate of UTI during 5 years after transplantation, with changes in macrophage marker ratio in the renal medulla and in the corticomedullary salt gradient, which may have an impact on the immune microenvironment of the graft and predispose it to UTIs [32] (Table 1).

UTIs in KT recipients are most common during the first 6 months after transplant [33], which may be explained by the increase of the factors mentioned above, such as surgical injury, urinary tract catheterization, recent hospitalization, and an increased state of immunosuppression.

## 3. Impact on Short- and Long-Term Outcomes

UTIs in KT recipients can complicate in acute pyelonephritis (APN) and sepsis, particularly in the early post-transplant period, increasing the risk of mortality [34,35]. However, it is unclear if UTIs have an impact on renal graft function. Some clinical studies have evaluated this association with divergent results. Some have found an association between a single episode of acute pyelonephritis and graft loss [36], others only in cases of recurrent UTIs [37], while others have not [25]. Britt et al. analyzed a cohort of 2469 renal transplant recipients and they found that UTIs, especially recurrent ones were significantly associated with decreased graft function, compared with those who did not have recurrent UTIs [37]. A review by Martin-Gandul et al. showed that the period of onset of the infection can impact graft dysfunction to varying degrees. They found that an early UTI is associated with the development of bacteremia and rejection, whereas late recurrent UTIs seem to be related to an increased risk of renal graft dysfunction and loss [38]. Abbott et al. documented that late UTIs were associated with an increased risk of death and graft loss [6]. Pellè et al. found that patients with APN exhibited both a significant increase in serum creatinine and a decrease in creatinine clearance, after 1 year and persisted 4 years after transplantation. Multivariate analysis revealed that APN represents an independent risk factor associated with the decline of renal function [25,34].

To summarize, the definitive effects of UTIs on kidney transplant patients are unknown. More studies are needed to evaluate the treatment of asymptomatic UTIs and prophylaxis protocols.

## 4. Etiology

The etiology of UTI in KT patients is similar to that of the general population, although the pathogens present a broader spectrum due to prior antibiotic treatments, hospitalizations, and urinary tract catheterization are more frequent in KT patients. UTIs after KT is usually caused by Gram-negative microorganisms (more than 70%) and *Escherichia coli* is the most common pathogen (30–80%) [34,39]. *Klebsiella, Proteus*, and *Pseudomonas aeruginosa* are other Gram-negative bacteria that are frequently isolated. On the other hand, Gram-positive pathogens (*Streptococcus*
*species*, *Staphylococcus saprophytic*) are less frequent [40].

The use of antibiotics in preventing infections or treatment of asymptomatic bacteriuria has led to a significant increase in resistance to common antibiotics and caused an increase of infections due to multidrug resistance (MDR) and extensively-drug-resistant (XDR) pathogens (*Enterococcus faecium, Staphylococcus aureus, Klebsiella pneumoniae, Acinetobacter baumannii, Pseudomonas aeruginosa,* and *Enterobacter*
*spp*) [41,42,43].

Other causes of UTIs in KT recipients with a different clinical presentation, different diagnostic procedures, and some of them specific to kidney transplantation or immunocompromised population are what we can call atypical ITUs. They include BK virus (BKV), cytomegalovirus (CMV), adenovirus (AdV), fungus, ureaplasma urealyticum, and tuberculosis (Table 2).

## 5. Classification

UTIs in the general population are classified based on the clinical presentation, the anatomical level, the grade of severity of the infection, the categorization of risk factors, and the availability of appropriate antimicrobial therapy [44]. The concepts of uncomplicated UTI and complicated UTI are well known, even though they are actually heterogeneous terms and current UTI guidelines use them with a number of modifications. The latest European Association of Urology (EAU) guidelines on urological infections of 2020 [41] define uncomplicated UTIs as acute, sporadic or recurrent lower and/or upper, limited to non-pregnant women with no known relevant anatomical and functional abnormalities within the urinary tract or comorbidities. Additionally, complicated UTI includes all UTIs not defined as uncomplicated, which is the case of all KT recipients. However, UTIs following kidney transplantation are classified by consensus in acute simple cystitis, acute pyelonephritis/complicated UTI, and recurrent UTI [2], due to their different clinical presentation, prognosis, and management.

Acute simple cystitis affects the lower urinary tract and consists of significant growth of auropathogen in urine culture with the presence of dysuria, urinary urgency or frequency or suprapubic pain without systemic symptoms such as fever, allograft pain or hemodynamic compromise, and no indwelling device such as ureteral stent, nephrostomy tube or chronic urinary catheter [2].Acute pyelonephritis/complicated UTI affect the upper urinary tract and include significant growth of a microorganism in urine culture with at least one of the following signs or symptoms: Fever, chills, malaise, hemodynamic instability, leukocytosis, and bacteremia caused by the same pathogen found in the urine or pain in the allograft or the costovertebral angles for allograft or native kidney involvement. Complicated UTIs also encompass severe syndromes including structural or functional abnormalities of the genitourinary tract (beyond normal transplantation reconstruction), involvement in associated organs such as prostatitis, and indwelling ureteric stents, bladder catheters or nephrostomy tubes [2].Recurrent UTI is defined as the occurrence of three or more UTIs in the last 12 months or two or more UTIs in the last 6 months [3]. It includes relapses and reinfections. A relapse is defined as the isolation of the same microorganism that caused the preceding infection in a urine culture obtained within 2 weeks after finishing the previous treatment, which means that the infection has persisted despite the treatment. Reinfection is a new episode of infection and occurs 2 weeks after the end of treatment or with a negative control urine culture, and can be due to the same or a different microorganism.

Asymptomatic bacteriuria (ASB) can be considered to be a separate entity apart from UTI (cystitis/pyelonephritis) since it is not necessarily a disease state [2]. ASB is defined as the presence of >10⁵ bacterial colony forming units per milliliter (CFU/mL) in urine culture without local or systemic signs and symptoms [4]. ASB is a common finding in KT recipients, occurring in the 17–51% of these patients [45], with a reported incidence of up to 50% during the first year after transplantation [46]. In the general population, ASB is also common and is considered to be a commensal colonization, with clinical studies that have shown that ASB may protect against superinfecting symptomatic UTI [47].

## 6. Diagnosis

Diagnosis of an UTI is based on the presence of lower urinary tract symptoms in cystitis (dysuria, urinary urgency or frequency or suprapubic pain) or upper urinary tract symptoms in pyelonephritis or complicated UTI (fever, chills, malaise, hemodynamic instability or allograft pain), and significant quantitative count of bacteria in an appropriately collected urine specimen [2] (Table 3). However, symptoms usually are masked due to immunosuppression and surgical denervation of the kidney allograft and ureter, for example, urinary symptoms and/or allograft pain cannot be present. Thus, the diagnosis should not only be based on classical signs and symptoms, since the clinical features frequently are not typical in this population. In fact, the first manifestation of UTI may be isolated fever or even a non-specific sepsis syndrome. Associated bacteremia is present in 3 to 16% of the cases of UTI [10,17], being more frequent in the first months after transplantation. Acute kidney failure is commonly observed in transplant pyelonephritis due to direct invasion of the kidney parenchyma by the uropathogen, resulting in acute infectious tubulointerstitial nephritis or associated with septic shock if it is present.

Urine analysis and urine culture are indicated if UTI is suspected, and blood cultures should be collected in case of severe systemic symptoms. In case of UTI, the urine dipstick is usually positive for nitrites, blood, protein, and leukocyte esterase, and the urine microscopy typically shows pyuria, defined by more than 10 white blood cells per milliliter (WBC/mL). Since the presence of pyuria alone cannot be used to diagnose a UTI, other different diagnoses should be considered in its absence. However, less than 10 WBC/mL in the urine may be indicative of a UTI if typical symptoms or a consistent clinical context are present. Bacteriuria is confirmed by a urine culture and allows for the determination of antibiotic sensitivity in order to direct the treatment, even retrospectively if the antibiotic has been started empirically. Bacteriuria is defined as the presence of bacterial growth in urine and is considered significant when it meets the standard quantitative criterion of greater than 10⁵ CFU/mL, in order to rule out contamination of the sample. However, this criterion varies according to the form of clinical presentation, 10³ CFU/mL for cystitis and 10⁴ CFU/mL for acute pyelonephritis or complicated UTIs are considered also significant growth [2]. It should be noted that the low CFU/mL in cystitis are mainly limited to *E. coli* [48], whereas not all organisms found in urine cultures are pathogens such as, *Staphylococcus epidermidis* (except in the presence of ureteral stents), *Lactobacillus*, and *Gardnerella vaginalis*. The urine can be contaminated during the collection of the sample by bacteria that colonize the distal urethra and genital mucosa, so it is recommended to clean the urinary meatus beforehand, collect the urine sample midstream in a sterile container, and in case it is not taken immediately to the laboratory, store in a refrigerator to avoid bacterial overgrowth. For the diagnosis of UTI, the sample must be collected before starting the antibiotic treatment and straight catheterization to obtain a urine specimen can be considered as an alternative. Furthermore, catheter removal and collecting either during midstream urination or via a newly placed catheter is recommended for patients with indwelling catheters, especially those in place for more than 2 weeks, and a suspected urinary tract infection [47]. The growth of multiple bacteria in a urine culture may indicate contamination of the sample or an incorrect collection technique.

Screening of ASB in KT patients has been controversial due to the supposed risk of progression to symptomatic UTI and the impact on graft survival. However, the majority of studies have not found this association and at present, there is no conclusive evidence that suggests screening for ASB and that its treatment improve outcomes. In fact, a positive effect of colonizing bacteria has also been proposed as in the general population, due to competition for nutrients or receptor sites against other more virulent microorganisms or to the host’s cross-protective immune or inflammatory response. Data suggest that ASB often spontaneously resolves itself without antibiotics. Moreover, the treatment of ASB has been associated with antibiotic harmful effects such as the development of multidrug-resistant (MDR) bacteria, a higher incidence of *Clostridium difficile* diarrhea, and increased healthcare costs [49]. Concern about ASB in the first 1–3 months after transplantation has been raised due to higher immunosuppression and surgical manipulation immediately post-transplant, however, there is not enough evidence to support this [50]. Thus, recent guidelines do not recommend routine screening and treatment of ASB in KT patients [2,3,47], however, screening can be considered in the first 1–3 months post-transplant, and treatment only if there have been two consecutive positive urine cultures for the same bacteria. Treatment of ASB of multi-drug resistant bacteria is not recommended. Nevertheless, treatment of persistent ASB can be considered in patients with associated, unexplained, impaired renal function due to the rare occurrence of asymptomatic pyelonephritis [2]. As in the general population, screening of ASB is indicated in pregnant women and prior to urological surgery, including before allograft biopsy, since in these cases, treatment is mandatory [2,47]. Despite the above, most transplant centers routinely screen for ASB.

In addition to laboratory testing, renal ultrasound or non-contrast computed tomography (CT) scan should be considered to assess complications such as obstruction and abscess, especially in patients with signs of severe infection or those who do not fully respond to initial therapy [51]. Ultrasonography is considered to be the primary imaging technique in the evaluation of kidney transplant [52]. The sonographic appearance of transplant infections is quite variable and nonspecific. The most common findings are diffuse, focal areas of increased or decreased parenchymal echogenicity with surrounding edema. Mucosal thickening or focal echogenicity in the pelvicaliceal system may suggest pyonephrosis or collecting system infections. The presence of gas within the parenchyma suggests emphysematous pyelonephritis. Finally, abscesses, which are not an uncommon finding, are characterized by complex fluid-filled cysts within the parenchyma [53]. Non-contrast CT scan is useful to identify nephrolithiasis, complex cyst, and other anatomical urinary abnormalities. CT-positron emission tomography (PET) scan should be indicated when infection of native kidneys cysts in patients with polycystic kidney disease is suspected, since the diagnosis can be difficult due to the large number of cysts of different densities that these patients present and the absence of specific radiological findings of infected cysts (Figure 1).

## 7. Atypical UTI in KT Patients

KT recipients are susceptible to a broad spectrum of infectious pathogens with frequently different signs and symptoms from immunologically normal individuals. In kidney transplant patients, a clear example are atypical UTI, where we can include in order of frequency: (1) BK virus (BKV), cytomegalovirus (CMV), and adenovirus (AdV), that produce tubulo-interstitial nephritis with acute renal failure while mimicking graft rejection and can cause graft loss [54,55,56], (2) fungus, the most common of which is Candida, that can cause urinary obstruction by fungus balls [57], (3) *Ureaplasma urealyticum* and tuberculosis, presenting with classic UTI signs and symptoms in which no bacteria grow via conventional culture methods [58,59]. Screening and early diagnostic methods are necessary to improve the prognosis of these infections that in the case of some of them can become fatal if they are not thought of and treated on time.

The period from 1 to 12 months or up to 1 year after transplantation (depending on how quickly immunosuppression has been tapered, the use of antilymphocyte “induction” therapy, and deployment of prophylaxis), reflect intensive immunosuppression with viral activation and higher risk of atypical UTI [57]. They can also appear after intensifying immunosuppression, for example, after graft rejection (Figure 2).

### 7.1. BK Virus

BKV is a nonenveloped, double-stranded DNA virus and a member of the *Polyomaviridae family*. Transmission is thought to involve respiratory and oral routes, and results in a seroprevalence of 82% in adulthood [60]. BKV is a latent infection, which can lie dormant in tissues most notably in the kidney [61]. After SOT, BKV infection can cause hemorrhagic cystitis, but in kidney transplant recipients produce ureteric stenosis and, more important and frequent, tubulointerstitial nephritis named BKV-associated nephropathy (BKVAN) that can cause acute or progressive renal failure, being one of the most important causes of premature graft loss. BKVAN within transplanted kidneys arises from either primary infection from the transplanted kidney itself or following reactivation from latency in the patient’s native urinary tract. BKV reactivation is induced by a relative or absolute immunodeficient status, so the median time to clinically apparent BKVAN is within the first year after transplantation or after treating an episode of rejection [54,62]. The incidence of BK viruria was reported to be 23 to 73%, that of BK viremia was 8 to 15%, and that of BKVAN was 1 to 7% [63].

Screening and early diagnostic methods are essential to improve the prognosis of BKV infection. The recommendation for screening for BKV is by way of quantitative DNA virus testing by polymerase chain reaction (PCR) of the urine every 1 to 3 months during the first 2 years after transplantation. However, BK viruria is sensitive to detecting active BKV infection but not specific for nephropathy and has a positive predictive value of 29 to 67%. Detection of BKV DNA in plasma may represent a better indicator for nephropathy, especially when the plasma BKV load is greater than 4 log¹⁰ copies/mL or duplicate testing 2 to 3 weeks apart, which has a positive predictive value greater than 90%. Whilst inter-laboratory standardization of such PCR assays is awaited, such discrete values remain subject to interpretation by individual centers, serological tests are usually of limited value. The gold standard for diagnosing BKVAN is kidney histology, including tubulointerstitial nephritis with cytopathic changes and positive immunohistochemistry using antibodies generally targeting cross-reacting SV40 large T-antigen or BKV antigens or in-situ hybridization for BKV nucleic acids [54,64].

The first step in the treatment of BKV infection is to reduce immunosuppressive drugs as soon as viremia is significant [54] although a balance is needed to avoid allograft rejection [62]. The use of mammalian target of rapamycin inhibitor (mTOR-i) (rapamune, everolimus) as a treatment option not only provides immunosuppression, reducing the risk of acute rejection, but also due to its behaviour as a metabolic pathway inhibitor for BKV it can also aid in the reduction in viral load, hence a lower risk of developing BKVAN [65]. TRANSFORM study [66], using everolimus as the main immunosuppressive regimen resulted in a lower incidence of CMV and BKV infections compared with the standard dose of calcineurin inhibitor (tacrolimus). The use of adjunctive antiviral therapies remains controversial. Some centers advocate the use of a low dose of cidofovir. Leflunomide, an immunosuppressant used in rheumatoid arthritis, and fluoroquinolones have some anti-BKV activity in vitro, but little efficacy in vivo. Repletion of serum immunoglobulins may be considered [54,57].

### 7.2. Cytomegalovirus (CMV)

CMV infection is the most important viral infection that can occur following SOT. The incidence is highly variable, depending on prophylaxis strategies, and can reach up to 60% among patients with CMV IgG Donor+/Recipient—. The incidence of CMV infection varies from 5 to 30% for patients with CMV IgG Recipient+, but the incidence can be as high as 50% in patients who received T-cell depletion therapy [67,68,69]. The median time to clinically apparent CMV infection is within the first year after transplantation [67].

CMV infection can directly and indirectly affect the kidney allograft. Direct effects include CMV syndrome (e.g., fever, fatigue, myalgia, and leucopenia) or tissue-invasive CMV diseases (e.g., gastritis, duodenitis, colitis, hepatitis, pneumonitis or nephritis). Kidney graft is a frequent target organ, both from subclinical and active infections [70]. In active infections, characteristic cytomegalic inclusions are identified in the tubular epithelium. CMV infection can be associated with acute or chronic graft dysfunction mainly due to tubulointerstitial nephritis, which may appear to be an allograft rejection. The incidence of CMV tubulointerstitial nephritis is much less frequent than BKV. On the other hand, CMV can predispose to the appearance of acute allograft rejection [67,69].

As with BKV infection, systematic screening and early diagnostic methods are essential to improve the prognosis of CMV infection. The recommendation for screening for CMV is by way of quantitative DNA virus testing of the whole blood using PCR every 1 to 3 months during the first 2 years after transplantation. A World Health Organization (WHO) International Reference Standard became available in 2010 from a clinical isolate (Merlin) with a titer of 5 × 10⁶ IU/mL. All laboratory tests should be calibrated to the WHO International Standard with results reported as IU/mL. Quantitative DNA is generally used for diagnosis, preemptive management strategies, and monitoring response to therapy. The highest viral loads often are associated with tissue-invasive disease and the lowest are associated with asymptomatic CMV infection [71]. Positive CMV cultures derived from respiratory secretions or urine are of little diagnostic value since many immunosuppressed patients secrete CMV in the absence of invasive disease. Serological tests are usually of limited value [57]. The gold standard for diagnosing CMV nephritis is kidney histology, including tubulointerstitial nephritis with cytopathic changes and positive immunohistochemistry or in situ hybridization for CMV nucleic acids [72].

Preventive strategies for CMV infection after solid organ transplants are either preemptive or prophylaxis treatment with valganciclovir. Patients with high risk (i.e., those who have D+/R— CMV IgG or who have received T-cell depletion for induction) should receive universal prophylaxis treatment, whereas patients with low to intermediate risk can undergo preemptive treatment (viral replication without clinical symptoms) [67].

The drugs of choice for CMV syndrome and tissue-invasive CMV disease are valganciclovir or intravenous ganciclovir, the two drugs have the same efficacy and similar long-term outcomes [73]. However, intravenous ganciclovir is preferred as the initial treatment for patients with severe or life-threatening CMV disease or those with gastrointestinal absorption problems. The treatment should be continued for a minimum of 2 weeks or until the clinical symptoms have resolved and the CMV viral load is below the lower limit of quantification on one or two consecutive weekly samples [67].

### 7.3. Adenovirus (AdV)

AdV is a double-stranded DNA virus with 52 different serotypes that infect humans and it is an important cause of infections in both immunocompetent and immunocompromised individuals [56,74]. In immunocompromised patients, persistent AdV infection can cause serious complications. In particular, AdV infection is one of the most severe viral infections in kidney transplant recipients and can cause graft loss due to the AdV-induced tubulointerstitial nephritis. The reported prevalence of AdV infection during the first year after kidney transplant is 11% by urine culture and 6.5% by serum PCR [75,76]. Other manifestations of urinary tract involvement may include hemorrhagic cystitis, ureteral obstruction with hydronephrosis, acute tubular necrosis or a mass lesion in the kidney [77]. Adenovirus interstitial nephritis is rare in kidney transplant recipients [77]. If AdV infection is donor-transmitted, disease onset can happen after a relatively short period (7 days) after kidney transplant [78].

A definitive diagnosis of AdV UTI was confirmed by positive results for the presence of AdV DNA in the urine using a qualitative polymerase chain reaction assay. Serological tests are usually of limited value [78,79]. The gold standard for diagnosing AdV nephritis is kidney histology [77].

For treatment of AdV UTI, reduction in immunosuppression is necessary, similar to other viral infections [80]. Gammaglobulin administration can be beneficial, particularly in cases of hypogammaglobulinemia. Antiviral agents such as cidofovir, acyclovir or ganciclovir should only be considered in cases that inadequately respond to a reduction in immunosuppression therapy and the administration of gammaglobulin [56].

### 7.4. Fungus-Candida

The most common fungal pathogen in transplant patients is Candida, with more than 50% being non-albicans species [57]. In kidney transplant recipients, candiduria is frequent but usually asymptomatic. However, in individuals with poor bladder function, obstructing fungal balls can develop at the ureteropelvic junction, resulting in obstructive uropathy, ascending pyelonephritis, and the possibility of systemic dissemination [57]. Candiduria has been reviewed in a Spanish consensus on urinary tract infection management in solid organ transplant recipients, and they have established the following recommendations [3]:Treatment of asymptomatic candiduria is not currently recommended for SOT recipients. Among patients with a urinary catheter, removal of the catheter may be sufficient to eliminate candiduria without specific antifungal therapy.Disseminated candidiasis should be considered in all hospitalized SOT with candiduria. If clinical manifestations are compatible, blood cultures, a second urine culture after removal or replacement of the urinary catheter, fundoscopy, cultures from any other significant site (vascular accesses, peritoneal fluid, etc.), and a kidney imaging study should be obtained.Patients with persistent candiduria and no indwelling bladder catheter should undergo imaging of the kidneys and collecting system to exclude renal abscess, fungus balls or other urologic abnormalities.Candida cystitis or pyelonephritis should be treated with systemic antifungals for 2–4 weeks.Fungus balls or casts in the pelvis or urinary bladder need surgery and systemic and/or local antifungal therapy.Fluconazole is the agent of choice for most patients with Candida UTI due to the high concentration achieved in urine.Other antifungal agents should only be considered for patients in unstable clinical condition, allergic to fluconazole or in whom therapy has clearly failed despite maximum fluconazole doses and optimal management of urologic abnormalities or other predisposing conditions.

### 7.5. Ureaplasma Urealyticum

*Ureaplasma urealyticum* is a small urease-hydrolyzing member of the order Mycoplasmatales which includes mycoplasmas and ureaplasmas as the medically-relevant organisms. Ureaplasma species commonly colonize the urogenital tract of women after the onset of sexual activity. Generally, in healthy individuals, these organisms are asymptomatic colonizers. Immunocompromised patients often present atypical or more invasive infections including intra-abdominal abscess. It is not a frequent infection in KT but it should be suspected in cases of signs and symptoms of classical UTI in whom no bacteria grow via conventional culture methods. *Ureaplasma urealyticum* requires a specific culture medium for optimal growth of mycoplasmas. Macrolides, doxycycline, and fluoroquinolones are the treatments of choice and they are intrinsically resistant to all beta-lactam antibiotics [58,81].

### 7.6. Tuberculosis

Urogenital tuberculosis (UGTB) is one of the great imitators, commonly masked by classical UTI causing a delay in diagnosis. It is the second to third most common form of extrapulmonary tuberculosis. UGTB is responsible for a destructive inflammation of the renal parenchyma and urinary tract, often leading to the loss of kidney function. The most common presenting symptoms are urinary irritation and lumbago. In the laboratory, we can find microscopic hematuria, sterile pyuria, and microscopic proteinuria [82]. In KT patients, tuberculosis increases in frequency and severity of disease with mortality rates 10-fold higher than in immunocompetent individuals. The most frequent source of tuberculosis infections in KTs is reactivation of quiescent foci of *Mycobacterium*, from the donor it represents approximately 4% of reported post-transplant tuberculosis cases. Much higher rates occur in endemic regions. Active disease should be excluded in PPD-positive living donors with chest radiograph, sputum cultures, and chest computed tomography if the chest radiograph is abnormal. Urine acid-fast bacillus cultures may be useful in a PPD-positive living kidney donor [57]. Recommendations for tuberculosis prophylaxis and treatment have been established by KDIGO clinical practice guideline for the care of kidney transplants recipients [60]. They suggest that tuberculosis prophylaxis and treatment regimens are the same in KT patients as would be used in the local, general population who require therapy. In addition, they recommend monitoring calcineurin inhibitor and mTOR-i blood levels in patients receiving rifampin and considering substituting rifabutin for rifampin to minimize interactions with the calcineurin inhibitor and mTOR-i.

## 8. Recurrent UTI

Recurrent UTIs after KT are defined as ≥2 UTI episodes in a 6-month period or ≥3 UTI episodes within 1 year. Recurrent UTIs occur in 3 to 27% of cases of KT recipients [2] [83]. The pathogenesis of recurrent UTI is multifactorial, underlying host factors, anatomical and functional abnormalities, and factors related to the KT all play different roles. The specific risk factors for recurrent UTI have not been well known. Many potential risk factors involved in the development of recurrent UTI have been described [83]. On the one hand, the female gender seems to be the strongest risk factor for recurrent UTI due to the shorter urethra and intercourse. Furthermore, immunosuppression, hyperglycemia, prostatitis, concomitant cytomegalovirus disease, and retransplantation are also associated with recurrent UTI [83]. On the other hand, anatomical abnormalities have been associated with recurrent UTI. The most important risk factor is the vesicoureteral reflux that occurs due to the disruption of the normal valve at the ureteric orifice as a consequence of the KT surgery [83,84]. To diagnose it, it is necessary to conduct a cystourethrography. In addition, urinary tract obstruction by renal calculi, bladder dysfunction, urethral abnormalities, and complex cysts have been also associated with recurrent UTI. Some diagnostic tests are necessary such as, radiography, ultrasonography, CT even CT-PET, cystoscopy, cystourethrography, and urodynamic studies [83,85]. Gram-negative organisms account for the majority of recurrent UTIs and *E. coli* remains the most frequent isolated uropathogen. Recently, the extended spectrum beta lactamase bacteria are the causative agent of recurrent UTI [83]. Recurrent UTIs have been associated with morbidity, increased mortality, and worse graft outcomes. Mortality rates are higher in patients with UTIs due to multidrug-resistant Gram-negative organisms [83]. Antibiotic selection should be guided by the results of microbiology tests given the emergence of drug-resistant uropathogens. The duration of antimicrobial therapy in recurrent UTIs is not well defined, but some authors recommend longer durations of treatment (4–6 weeks or even longer) [4,83]. Prevention of recurrent UTIs in KT patients has not been properly researched. Currently, therapies tend to have some overlap with strategies to prevent recurrent UTIs in the non-transplant population [4]. If no anatomic reason is found, lifestyle modification and medical prophylaxis should be initiated. Behavioral education is the first tool to decrease recurrent UTI, such as hydration, frequent voiding, and intimate hygiene, especially for females after a sexual intercourse [2]. The use of antibiotic prophylaxis for the prevention of recurrent UTI has not been well studied. The potential benefits of extended antibiotic prophylaxis should be carefully weighed against the risks of promoting bacterial resistance, *Clostridium difficile* infection, and other adverse events associated with antibiotics. Antimicrobial prophylaxis might be appropriate for selected patients who have severe episodes of recurrent UTIs such as pyelonephritis [86].

Nonantimicrobial therapies have been tried in patients with recurrent UTI most of all in the non-transplant population. It has been described that cranberry juice could prevent the adhesion of bacteria to the uroepithelium, similarly topical estrogen has been used in perimenopausal and postmenopausal females and a combination of hyaluronic acid and chondroitin sulphate instilled intravesically. Other therapies as methenamine and the use of probiotics are well tolerated and may be useful in reducing recurrent UTIs in KT recipients [86,87]. Finally, it has recently been published that bacterial vaccines reduced the incidence of recurrent UTI in a small case series of KT patients without eliciting any safety concerns [88]. Despite these results, these therapies should be researched further in KT patients with recurrent UTIs.

## 9. Treatment of UTIs in KT Recipients

Some special considerations must be made in the treatment of UTIs in KT recipients. First, collecting a urine culture prior to empiric antibiotic treatment is mandatory in order to switch to the narrowest spectrum antibiotic available to complete the course of therapy, once culture susceptibility results are available. Second, the treatment strategy and its duration depend on the time elapsed since transplantation and the severity of the illness. Third, the antibiotic dosage should be adjusted according to the patient’s renal function and nephrotoxic antibiotics may be avoided if it is possible. Interactions between antimicrobial and immunosuppressive drugs make the treatment more complex since the co-administration of them can modify the pharmacokinetic and pharmacodynamic characteristics of both groups of drugs, causing serious consequences. Therefore, recognizing these interactions, monitoring plasma drug levels, as well as, renal function are essential [3]. Finally, the option of reducing/discontinuing immunosuppression therapy may be considered in the case of severe infection with sepsis, and sometimes in the case of recurrent UTIs [3]. Usually, the immunosuppressive drug that is initially reduced or discontinued is purine inhibitors (azatioprine or mycophenolate mofetil). Another important issue is to consider removal or replacement of urinary tract instruments such as urethral catheters or urologic stents [3].

The choice of empirical antimicrobial agents should be based on local epidemiological data, the patient’s history of previous resistant organisms, and antibiotic therapies prescribed in the previous months. In general, the empirical antibiotic therapy which can be recommended is:Simple cystitis: Fosfomycin 3 g (two doses) or amoxicillin/clavulanate or second/third generation oral cephalosporins [3].Alternative therapy: TMP/SMX or ciprofloxacin. Duration: 7–10 days in the first 6 months post-transplant, 5–7 days beyond 6 months [2].Acute pyelonephritis/complicated UTI: Piperacillin-tazobactam or cefepime or carbapenem, ± fluoroquinolone [2].Duration: 14–21 days of therapy with the most narrow-spectrum antibiotic available [2].

## 10. Summary

UTIs are the most common infection in kidney transplant patients, and are a major cause of morbidity, hospitalization, and mortality, with impact on the patient’s well-being. In addition, they are associated with other complications such as potential effects on graft survival. For this reason, early diagnosis and adequate antibiotic treatment are very important. However, in the past, urine cultures were routinely performed at each transplant patient visit and treated even if they were asymptomatic. This led to the emergence of multi-resistant pathogens and its effectiveness has not been proven except, maybe, in the first months after transplantation. The antibiotics used in KT patients are the same as in the general population, avoiding nephrotoxic drugs, and the duration of treatment should be longer. Finally, it is important to think about other pathogens that can cause atypical urinary tract infections such as virus, fungus, and tuberculosis.

## Figures and Tables

**Figure 1 diagnostics-11-01456-f001:**
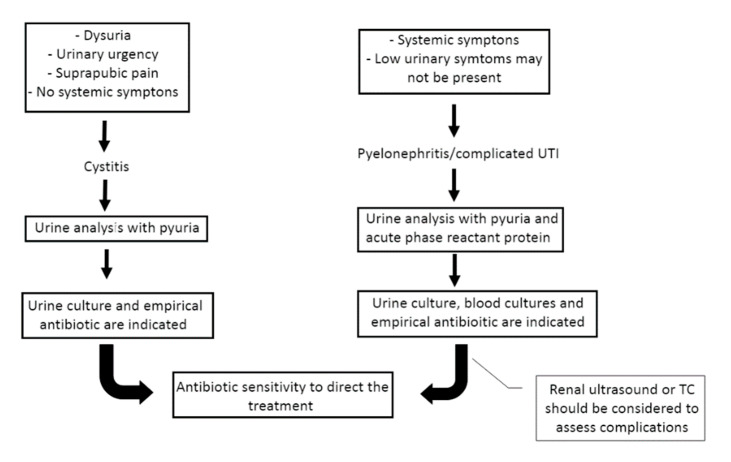
Diagnosis of typical urinary tract infection.

**Figure 2 diagnostics-11-01456-f002:**
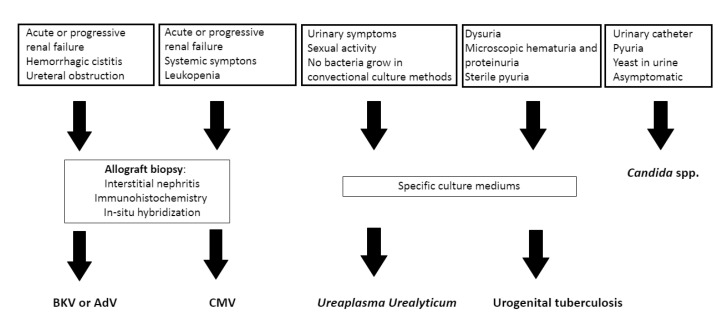
Diagnosis of atypical urinary tract infection.

**Table 1 diagnostics-11-01456-t001:** Risk factors to urinary tract infection (UTI) in kidney transplant recipients.

Pretransplant	Graft Related	Post-Transplant
Female gender	Deceased donor	Vesicoureteric reflux
Older age	Expand criteria	Ureteral stent (>14–21 days)
Diabetes mellitus	Infected donor organ	Prolonged bladder catheterization
Urinary tract abnormalities	Contaminated graft perfusion solution	Delayed graft function
Lower urinary tract dysfunction		Bladder dysfunction
History of recurrent UTI		Immunosuppression (Thymoglobulin, mycophenolate)
Polycystic kidney disease		Long hospitalization
Time in dialysis		Episodes of acute rejection
Disused bladder		Diabetes mellitus post-transplantation

**Table 2 diagnostics-11-01456-t002:** Microorganisms causing UTI in KT recipients.

Typical UTI		Atypical UTI
More Frequent	Less Frequent	
*Escherichia coli*	*Staphylococcus saprophyticus*	BK virus
*Klebsiella* spp.	*Streptococcus agalactiae*	Cytomegalovirus
*Enterococcus* spp.	*Staphylococcus aureus*	Adenovirus
*Proteus* spp.	*Citrobacter* spp.	Candida spp.
*Pseudomonas aeruginosa*	*Enterobacter* spp.	Tuberculosis
	*Morganella* spp.	*Ureaplasma* spp.
	*Providencia* spp.	*Corynebacterium urealyticum*
	*Serratia* spp.	

**Table 3 diagnostics-11-01456-t003:** Classification of urinary tract infection (UTI) in renal transplant recipients.

	Simple Cystitis	Acute Pyelonephritis/ Complicated UTI	Recurrent UTIs
Urine culture	>10³ bacteria CFU/mL.	>10⁴ bacteria CFU/mL.	>10³ or 10⁴ bacteria CFU/mL according to the type of UTI.
Clinical presentation	Dysuria, urinary urgency and/or frequency or suprapubic pain. No systemic symptoms and no indwelling urinary catheters.	Fever, chills, malaise, hemodynamic instability; flank/allograft pain or bacteremia with the same organism as in urine. Abnormalities of the genitourinary tract and/or indwelling urinary catheters. Low urinary symptoms may or may not be present.	≥3 UTIs in prior 12-month period or ≥2 UTIs in the last 6 months.
Treatment	Outpatient treatment. 7–10 days in the first 6 months post-transplant. 5–7 days beyond 6 months.	Hospitalization is required for 14–21 days. In severe infection, reduction/discontinuation of immunosuppression should be considered.	Longer time of treatment (4–6 weeks) and lower dose of prophylaxis after. Evaluation of possible causes. Non-antimicrobial prevention strategies.

## Data Availability

Not applicable.

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
