# Peer review of "A Current Review of the Etiology, Clinical Features, and Diagnosis of Urinary Tract Infection in Renal Transplant Patients"

_diagnostics, 2021, doi:10.3390/diagnostics11081456_

Round 1

Reviewer 1 Report

The topic of this manuscript, urinary tract infections (UTIs) in kidney transplant (KT) patient is an important disease, and I am interested in this topic.

Etiology, clinical features and diagnosis of UTIs in KT patients are well summarized in this manuscript.

This review is worth reading by many clinicians. However, the following revisions are required.

[Major revisions]

  1. Atypical UTIs in KT patients are summarized in this manuscript, but typical UTIs are more important. The authors should summarize typical UTIs in KT patients, especially major pathogens that cause UTIs in KT patients.
  2. The authors should summarize treatments of UTIs in KT patients, especially differences from that of UTIs in ordinary patients

[Minor revisions]

Page 1, Title: The width of the space between “the” and “etiology” is strange.

Author Response

We appreciate your comments and revisions. Our responses are the following:

  1. Atypical UTIs in KT patients are summarized in this manuscript, but typical UTIs are more important. The authors should summarize typical UTIs in KT patients, especially major pathogens that cause UTIs in KT patients.

Response 1: We have summarized atypical UTIs in KT patients individually because they are specific to KT and their clinical presentation differs from typical UTIs and they vary from each other. However, we have also briefly summarized the most frequent microorganisms in typical UTIs in the paragraph “Etiology”, because these microorganisms are the same as in the general population. In order to clarify this issue, we can add a sentence in this paragraph and a table with the different pathogens, as we have proposed in the Word uploaded.    

2. The authors should summarize treatments of UTIs in KT patients, especially differences from that of UTIs in ordinary patients.

Response 2: we have not included a section about treatment because the idea of the article was about diagnosis, but I agree that we can incorporate a paragraph about treatment at the end of the article.

  1. Page 1, Title: The width of the space between “the” and “etiology” is strange.

Response 3: you are correct, it is a mistake.

Please see the attachment. We have also included the revisions of the other reviewer.

Reviewer 2 Report

The review article by Suárez et al is an interesting, well written up to date review on the field. However, there are some issues that have to be solved in the manuscript. One of the critical issues is that the review finishes abruptly without providing new possible options for treatment or discussion concerning the use of different immune suppressive therapy in different types of patients. For example, doi: 10.3389/fphar.2021.635165 does an anaylisis of pharmacokinetics of calcineurin use in elderly population 

Also, this important article is missing doi 10.3390/v13030487. Why did the authors did not discuss diabetes after transplant as a risk factor? 

  • 10.3390/medicina57030250 It would be important.
  • Finally, I would suggest the authors to elaborate a flowchart for the diagnosis of typical and atypical conditions of UTI. 

Author Response

We appreciate your comments and revisions. Our responses are the following:

  1. One of the critical issues is that the review finishes abruptly without providing new possible options for treatment or discussion concerning the use of different immune suppressive therapy in different types of patients. For example, doi: 10.3389/fphar.2021.635165 does an anaylisis of pharmacokinetics of calcineurin use in elderly population.  

Reponse 1: I agree with your comment, so we have added a paragraph with a summary of the article to finish. We have also included a new paragraph about special considerations in the treatment of UTI in KT recipients, including management of immunosuppressive therapy, although we have not delved into the subject.  

  1. this important article is missing doi 10.3390/v13030487.

Response 2: We have added this reference.

  1. Why did the authors did not discuss diabetes after transplant as a risk factor?  10.3390/medicina57030250 It would be important.

Response 3: We have not discussed diabetes after transplant (DMPT) because the risk is the same than diabetes mellitus pretransplant or in the general population, so we do not considered it as a specific risk factor in KT recipients. We can include DMPT in the table 1 in postrasplant risk factors. However, we have added SGLT2 inhibitors as a risk factor of UTI. 

  1. Finally, I would suggest the authors to elaborate a flowchart for the diagnosis of typical and atypical conditions of UTI. 

Response 4. We have included two figures about diagnosis to typical and atypical UTI.

Please, see the attachment. We also included our revisions of the other reviewer. 

Round 2

Reviewer 1 Report

I have reviewed the revised version.

The manuscript has been significantly improved.

I recommend that the manuscript be accepted for publication in Diagnostics.